# RNA Sequencing of H3N2 Influenza Virus-Infected Human Nasal Epithelial Cells from Multiple Subjects Reveals Molecular Pathways Associated with Tissue Injury and Complications

**DOI:** 10.3390/cells8090986

**Published:** 2019-08-27

**Authors:** Kai Sen Tan, Anand Kumar Andiappan, Bernett Lee, Yan Yan, Jing Liu, See Aik Tang, Josephine Lum, Ting Ting He, Yew Kwang Ong, Mark Thong, Hui Fang Lim, Hyung Won Choi, Olaf Rotzschke, Vincent T Chow, De Yun Wang

**Affiliations:** 1Department of Otolaryngology, Yong Loo Lin School of Medicine, National University of Singapore, Singapore 119228, Singapore; 2Singapore Immunology Network (SIgN), A*STAR, Singapore 138648, Singapore; 3Center for Interventional Medicine, The Fifth Affiliated Hospital of Sun Yat-sen University, Zhuhai 519000, China; 4Division of Respiratory and Critical Care Medicine, National University Hospital, Singapore 119074, Singapore; 5Department of Medicine, Yong Loo Lin School of Medicine, National University of Singapore, Singapore 117597, Singapore; 6Institute of Molecular and Cell Biology, A*STAR, Singapore 138673, Singapore; 7Department of Microbiology and Immunology, Yong Loo Lin School of Medicine, National University of Singapore, Singapore 117545, Singapore

**Keywords:** influenza, H3N2 virus, human nasal epithelial cells (hNECs), RNA sequencing, transcriptomics, epithelial responses, pathogenesis

## Abstract

The human nasal epithelium is the primary site of exposure to influenza virus, the initiator of host responses to influenza and the resultant pathologies. Influenza virus may cause serious respiratory infection resulting in major complications, as well as severe impairment of the airways. Here, we elucidated the global transcriptomic changes during H3N2 infection of human nasal epithelial cells from multiple individuals. Using RNA sequencing, we characterized the differentially-expressed genes and pathways associated with changes occurring at the nasal epithelium following infection. We used in vitro differentiated human nasal epithelial cell culture model derived from seven different donors who had no concurrent history of viral infections. Statistical analysis highlighted strong transcriptomic signatures significantly associated with 24 and 48 h after infection, but not at the earlier 8-h time point. In particular, we found that the influenza infection induced in the nasal epithelium early and altered responses in interferon gamma signaling, B-cell signaling, apoptosis, necrosis, smooth muscle proliferation, and metabolic alterations. These molecular events initiated at the infected nasal epithelium may potentially adversely impact the airway, and thus the genes we identified could serve as potential diagnostic biomarkers or therapeutic targets for influenza infection and associated disease management.

## 1. Introduction

The global burden of inter-pandemic influenza is high. It is estimated to affect 1 billion people annually, with 3–5 million severe cases requiring hospitalization or intensive care treatment, resulting in approximately 0.5 million deaths [1]. Worryingly, drug-resistant influenza strains are emerging at a rapid rate that will severely hamper the ability of our healthcare systems to contain influenza outbreaks [2]. Therefore, alternative strategies are needed against severe influenza infections during both seasonal and pandemic influenza outbreaks.

The normal human airway epithelium is a pseudo-stratified layer of ciliated and non-ciliated columnar cells, goblet cells, club cells, and basal cells [3]. The airway epithelium protects against airway infection via efficient mucociliary clearance (MCC), the production of inflammatory mediators and chemokines against viruses, and the recruitment of immune cells [4]. When the influenza virus breaches the defense of the human airway epithelium, it causes a myriad of innate responses by the infected host in response to viral invasion [5,6]. Among these changes are critical factors that can determine disease severity, and which may lead to the development of diagnostic, prognostic prediction markers, or anti-influenza therapies [7,8,9,10]. However, few studies have hitherto been performed in relevant models [11] and human models of influenza are not feasible due to potential severity of the infection. Therefore, the mechanistic study of viral-induced airway changes using relevant models can lead to better understanding of the development of severe complications. Additionally, we need greater clarity on the different immune responses in view of the rising prevalence of chronic diseases such as diabetes mellitus and asthma. Patients with these disorders are especially susceptible to severe influenza complications compared to healthy subjects [12]. Thus, the establishment of a baseline response against influenza infection of “healthy” tissue is beneficial to facilitate future comparative studies to better manage influenza in patients with co-morbidities.

Although the study of host responses in influenza infection is not new, current in vitro cell lines cannot accurately represent human airway infection due to the lack of key mucociliary features [13]. Hence, we have previously developed an air–liquid interface (ALI) human airway epithelial cell culture that is able to sustain influenza infection [5,6]. We have also further compared the transcriptomic responses of our infected human nasal epithelial cells (hNECs) with 15 other in vitro and in vivo influenza infection transcriptomic studies [6]. The comparison revealed that at their peak responses against influenza, the differential transcriptome signature in hNECs was highly similar to the signatures from other influenza infection models [6]. Interestingly, compared to the homogenous cell lines tested, our heterogenous hNEC model exhibited a more comparable response to the clinical influenza studies, indicating that most responses were initiated at the nasal epithelium [6]. Therefore, in this study, we aim to further utilize the hNEC model as a physiologically relevant in vitro model to clarify the nasal epithelial responses against influenza H3N2 infection, which would then facilitate the identification of the key host factors that are significant for future studies.

To establish host factors that are significantly altered in the nasal epithelium as a reference of early innate responses against influenza, the dynamic expression of the genes needs to be clearly elucidated. While there are many studies that utilize microarray analysis to identify the host responses against influenza, the limitation of the microarray is its inability to determine the full extent of gene changes due to its hybridization-based protocol [14]. The aim of this study was to utilize RNA sequencing (RNAseq) technology to not only reveal the hNEC responses (from multiple individuals) against influenza infection, but also to identify those genes with high magnitude changes to serve as potential reference markers of the innate responses of influenza infection. Given that RNAseq functions by reading virtually all the RNAs present in the samples tested, we can also discern the magnitude of each RNA change and mark them as the canonical responses. In addition, as RNAseq is not constrained by probe usage as in microarrays, they are therefore more reliable in detecting novel interactions during influenza infections of hNECs. Hence, RNAseq analysis will further augment the transcriptomic data established previously by microarray analysis. The augmented baseline can then be applied to future clinical studies and practice against influenza infection, especially for comparison against patients with other underlying co-morbidities that may be affected by more severe disease.

## 2. Materials and Methods

### 2.1. Derivation of Human Nasal Epithelial Stem/Progenitor Cells (hNESPCs) and In Vitro Differentiation of hNECs

Approval to conduct this study was obtained from the National Healthcare Group Domain-Specific Board of Singapore (DSRB Ref: D/11/228) and the institutional review board of the National University of Singapore (IRB Ref: 13-509). Written consent was obtained from donors prior to the collection of the tissue biopsies. At the time of collection, all subjects were free of symptoms of URTI. The medical backgrounds of the subjects are summarized inAppendix A. The hNESPCs were isolated and enriched from the tissue biopsies according to a previously standardized protocol [5,15], which normalized the hNESPCs to a baseline state that differentiates into hNECs resembling healthy tissues if they pass the quality control checks for their differentiation [5]. Following enrichment, the hNESPCs were expanded further and subjected to ALI culture in Transwells for in vitro differentiation according to previous protocol as well [5,15]. Briefly, primary cells were subjected to isolation for selection of hNESPCs, which were enriched and expanded with Dulbecco’s Modified Eagle Medium: Nutrient Mixture F-12 (DMEM/F12) (Gibco-Invitrogen, Carlsbad, CA, USA) containing 10 ng/mL of human epithelial growth factor (EGF, Gibco-Invitrogen, Carlsbad, CA, USA), 5 μg/mL of insulin (Sigma, St. Louis, MO, USA), 0.1 nM of cholera toxin (Sigma, St. Louis, MO, USA), 0.5 μg/mL of hydrocortisone (Sigma, St. Louis, MO, USA), 2 nM/mL of 3,3′,5-triiodo-l-thyronine (T3) (Sigma, St. Louis, MO, USA), 10 μL/mL of an N-2 supplement (Gibco-Invitrogen) and 100 IU/mL of antibiotic-Antimycotic (Gibco-Invitrogen, Carlsbad, CA, USA). The expanded hNESPCs were then transferred onto 12-well 0.4 μm Transwell inserts (Corning, Corning, NY, USA). Once confluent, growth medium was discarded and 700 μL of PneumaCult™-ALI Medium with inducer supplements (STEMCELL Technologies Inc., Vancouver, Canada) was added to the basal chamber to establish ALI conditions. The cells were cultured in ALI culture for 4 weeks, with media change every 2–3 days. After 3–4 weeks of differentiation, hNECs from a total of seven donors were then subjected to influenza H3N2 virus infection.

### 2.2. Inoculation of Human Influenza H3N2 Virus in Fully Differentiated hNECs and Viral Quantification

The Influenza A strain used in this study is of the H3N2 subtype (A/Aichi/2/1968) (ATCC, Manassas, VA, USA). The virus was propagated using embryonated egg culture and used for all the infection in the hNECs. Prior to infection, fully differentiated hNECs were washed with 1× dPBS and infected with the H3N2 influenza virus at a multiplicity of infection (MOI) of 0.1 and incubated for 1 h at 35 °C. After the 1 h incubation, the viral inoculum was removed and the hNECs were incubated back in 35 °C. The control hNECs were harvested for apical wash and RNA prior to the infection at 0 h post-infection (hpi). The infected hNECs were then harvested for the apical wash and RNA following 8, 24, and 48 hpi incubation at 35 °C.

### 2.3. Viral Plaque Assay

At each infection time point, 150 µL of 1x dPBS was added and incubated in the apical chamber for 10 min at 35 °C to recover progeny viruses as the apical wash. The plaque assay for viral quantification was performed using overnight MDCK cultures (ATCC, Manassas, VA, USA) at 85–95% confluence in 24-well plates. The MDCK cells were incubated with 100 µL of serial dilutions (from 10^−1^ to 10^−6^) of virus from apical washes at 35 °C for 1 h, where plates were rocked every 15 min to ensure equal viral distribution. After incubation, the inocula were removed and replaced with 1 mL of Avicel (FMC Biopolymer, Philadelphia, PA, USA) overlay, and incubated at 35 °C for 65–72 h. After incubation, Avicel overlay were removed, and cells were fixed with 4% formaldehyde in 1× PBS for 1 h. Formaldehyde was then removed, and cells were washed with 1× PBS prior to staining with 1% crystal violet for 15 min before washing the stain away. The plaque-forming units (PFU) were calculated as follows: Number of plaques × dilution factor = number of PFU per 100 µL.

### 2.4. Total RNA Extraction and Real-Time Quantitative PCR

At each time point after the collection of apical wash, the hNECs were lysed using RNA lysis buffer. Total RNA was then extracted from the lysate using mirVana miRNA isolation kit (Life Technologies, Grand Island, NY, USA). The extracted total RNA was first subjected to nanodrop analysis to first ensure the RNA quality, before being submitted for RNAseq analysis. Then, 500 ng from the remaining RNA was subjected to cDNA synthesis using Maxima first-strand cDNA synthesis kit (ThermoScientific, Pittsburgh PA, USA). After this, qPCR analysis was performed to evaluate the transcriptional levels of host response genes selected based on previous microarray analysis using pre-designed primers (Sigma Aldrich). Each qPCR reaction was performed in duplicate using GoTaq-qPCR Master Mix kit (Promega, San Luis Obispo, CA, USA), and relative gene expression was calculated using the comparative method of 2-ΔΔCt normalized to the housekeeping gene PGK1. Relative gene expression levels were presented as median values and interquartile ranges, while statistical significance was determined using the Wilcoxon signed-rank test.

### 2.5. Library Preparation for RNAseq

All human RNAs were analyzed on the Agilent Bioanalyzer (Agilent, Santa Clara, CA, USA) or the Perkin Elmer Labchip GX system (Perkin Elmer, Waltham, MA, USA) for quality assessment with RNA Integrity Number (RIN) or RNA Quality score range from 6.8–9.7 and median of 9.0. cDNA libraries were prepared using 2 ng of total RNA and 1 µL of a 1:50,000 dilution of ERCC RNA Spike in Controls (Ambion^®^ Thermo Fisher Scientific, Waltham, MA, USA) using SMARTSeq v2 protocol [16], except for the following modifications: (1) Use of 20 µM TSO, and (2) use of 250 pg of cDNA with 1/5 reaction of Illumina Nextera XT kit (Illumina, San Diego, CA, USA). The length distribution of the cDNA libraries was monitored using DNA High Sensitivity Reagent Kit on the Perkin Elmer Labchip GX system (Perkin Elmer, Waltham, MA, USA). Sixteen samples were subjected to an indexed PE sequencing run of 2 × 51 cycles on an Illumina HiSeq 2000 (16 samples/lane), and 65 samples to an indexed PE sequencing run of 2 ×151 cycles on an Illumina HiSeq 4000 (30 samples/lane).

### 2.6. RNAseq Analysis

FASTQ files were mapped to the human genome build hg38 using STAR. Gene counts were computed using featureCounts (part of the Subread package) using annotations from GENCODE version 26. Differential gene expression analysis was performed using edgeR in a paired fashion under R version 3.3.3. Multiple testing correction was done using the method of Benjamini and Hochberg and *p*-values (False Discovery Rate; FDR) less than 0.05 was deemed to be significant.

### 2.7. Geneset Enrichment Analysis

Geneset enrichment analysis using data from Gene Ontology (GO) was performed using the Bioconductor package topGO, while the analysis using Reactome Pathway was performed using the Vioconductor package ReactomePA. Both analyses were run in R version 3.3.3 using multiple testing-corrected significant differentially-expressed genes.

### 2.8. Luminex Assay for Cytokines and Chemokines

Luminex assay was performed on apical supernatant collected from both uninfected control and infected hNECs at 8, 24, and 48 hpi using the Multiplexing Immunoassay Human Cytokine/Chemokine Panel 1, 41-plex (Merck, cat# HCYTOMAG-60K-41, Kenilworth, NJ, USA). The panel included the detection of: Human sCD40L, EGF, FGF-2, Flt-3 ligand, Fractalkine, G-CSF, GM-CSF, GRO, IFN-α2, IFN-γ, IL-1α, IL-1β, IL-1ra, IL-2, IL-3, IL-4, IL-5, IL-6, IL-7, IL-8, IL-9, IL-10, IL-12 (p40), IL-12 (p70), IL-13, IL-15, IL-17A, IP-10, MCP-1, MCP-3, MDC (CCL22), MIP-1α, MIP-1β, PDGF-AB/BB, RANTES, TGF-α, TNF-α, TNF-β, VEGF, Eotaxin/CCL11, and PDGF-AA. Samples and standards were incubated with fluorescent-coded magnetic beads which had been pre-coated with the respective capture antibodies. After an overnight incubation at 4 °C, plates were washed twice. Biotinylated detection antibodies were incubated with the complex for 1 h, and Streptavidin-PE was then added and incubated for another 30 min. Plates were washed twice again, then beads were re-suspended with sheath fluid before acquiring on the FLEXMAP^®^ 3D (Luminex) using xPONENT^®^ 4.0 (Luminex) acquisition software. Data analysis was done on Bio-Plex Manager™ 6.1.1 (Bio-Rad). Standard curves were generated with a 5-PL (5-parameter logistic) algorithm, reporting values for both MFI and concentration data. Results were then expressed as mean fold change compared with uninfected control, and *p*-values (FDR) of less than 0.05 were considered significant.

## 3. Results

### 3.1. Influenza Virus Infection Induces Similar Responses in hNECs from Multiple Donors

Prior to analysis, the responses of all seven hNECs donors following influenza infection were plotted on a principal component analysis (PCA) plot. The PCA plot indicated a degree of variability in the responses between donors and time points (Figure 1). Nonetheless, the responses were clustered tightly enough following infection to signify their consistency of infection for further transcriptomic analysis—similar to those observed in our previous study [6].

### 3.2. Transcriptomic Changes Are Detected at 8 hpi, and Peak at 48 hpi

Significant gene expression changes (FDR < 0.05) of infected hNECs were detected as early as 8 hpi, and further increased at 24 and 48 hpi (Table 1; Figure 2A). Also, the number of genes decreased in a linear fashion as the fold change in expression increased, as seen in the 10x fold change genes indicated in Figure 2B, where about 10% of the significantly altered genes remained. At 8 hpi, there were 31 upregulated genes and 13 downregulated genes. The major upregulated genes were the antiviral sensors and early response genes such as *IFN*s, *IFIT*s, and *IFI*s. Interestingly enough, interferon lambda (IFNλ) gene *IFNL*s was the earliest response interferon of infected hNECs, as opposed to interferons alpha or beta, at 8 hpi. At later time points, the number of gene expression changes increased substantially, with upregulation of 704 and 1080 genes, and downregulation of 217 and 758 genes at 24 and 48 hpi, respectively. There was augmented expression of antiviral effectors and inflammatory genes at both time points. *IFNL* remained the interferon gene with highest expression at both time points, while a marked elevation of cytokines such as *CXCL10* and *CXCL11* was also observed. Considering downregulated genes, proliferative and transcriptomic functions appeared to be suppressed, with diminished expression of genes such as *FMO2*, *KLK12*, and *FOSB*. Genes associated with metabolism, cell cycle, and DNA repair were further suppressed following infection at 24 and 48 hpi. Appendix A list the complete set of significant gene expression changes, arranged according to their fold change (Log_2_FC). In addition, we have also verified that the genes showing major expression changes by RNAseq generally concurred with RT-qPCR analyses. Of the 10 genes tested by qPCR at 48 hpi, all of them showed the same directional changes in expression as observed by RNAseq. Hence, seven of these genes showed a *p*-value of <0.05 (*IL4I1*, *IFNL1*(*IL29*), *CXCL10, TNFSF10*, *IFI6*, *CCL24*, and *CYP26A1*), one gene had a *p*-value of <0.1 (*CTGF*), while only two genes were not statistically significant (*TGFA* and *ANO5*) (Appendix A).

### 3.3. Transcriptomic Change Alterations Correlate with Viral Titer, with Intersecting Genes Remaining Consistent in Directional Changes of Expression

We then further compared the transcriptomic alterations in the hNECs over time, following influenza H3N2 infection. The number of gene expression changes mirrored the viral titer changes, which peaked at 48 hpi, and were consistent between donors (Figure 3A). Approximately two thirds of genes at 8 and 24 hpi overlapped with other time points, while about one third of genes at 48 hpi overlapped (Figure 3B). The overlapping genes displayed similar directional consistency at the significant time points. In addition, congruent with the consistent viral titer with most gene expression changes at 48 hpi, we also noted the most consistent alterations in expression of genes across donors. This is highlighted in Figure 3C, which portrays the heatmaps of the top 10 genes with the smallest *p*-value, together with their direction and magnitude of change. Based on these analyses, we proposed that 48 hpi represents the optimal time point for the subsequent pathway analysis to ascertain influenza-specific pathway changes.

### 3.4. Pathway Enrichment of Significant Gene Changes Revealed Potential Epithelium-Initiated Pathways Contributing to Influenza Pathology and Pathogenesis

We then further subjected the significant gene changes to gene set enrichment using both GO and REACTOME databases. At time points 8, 24, and 48 hpi, there were 3, 41, and 30 significant (adjusted *p*-value < 0.05) GO biological processes (Appendix A) and 3, 93, and 74 significant (adjusted *p*-value < 0.05) REACTOME pathways (Appendix A), respectively. At the early time of 8 hpi, interferon-mediated antiviral responses were elevated as expected. At 48 hpi, the pathways appeared to be more stabilized and consistent for both GO and REACTOME analyses, despite displaying more gene expression changes at this time point.

Responses to influenza virus skewing towards type I immunity were predominant in the GO analysis. The expected interferon-mediated functions by the epithelium validated the authenticity of our model, where we found enriched type I interferon (GO) and RIG-I (REACTOME) pathways with upregulation of nearly all significant gene members (data not shown). Besides the interferon and antiviral pathways, we identified several functions of interest initiated by the nasal epithelium that may contribute to the pathology and pathogenesis of influenza. At 48 hpi, GO pathway enrichment analysis revealed that the nasal epithelium was actively involved in initial IFNγ signaling (GO:0060333), despite not directly producing IFNγ. We also observed enriched function in apoptosis and necroptosis (GO:0008637 and GO:0060544), immune evasion (GO:0045824), and other pathways that may lead to complication events such as smooth muscle proliferation (GO:0048661) and response to fatty acid (GO:0070542) (Table 2).

For the REACTOME pathway analysis, we selected pathways that were enriched with more than 30 significant genes present in the enriched pathway, and these were generally in agreement with the GO analysis (Table 3). In addition to IFNγ signaling (877300) and apoptosis (109581), it also revealed changes in epithelial-initiated B cell receptor signaling (983705 and 1168372) and amino acid metabolism (71291) following influenza infection. It is noteworthy that these pathways were initiated at the epithelial level without the participation of immune cells, thus highlighting the relevant genes of interest for future studies.

### 3.5. Comparison with Established Influenza Gene Expression Signatures and Transcriptomes Revealed Consistent Upregulation of Influenza Response Genes

Given that RNAseq analysis facilitates more accurate expression changes following infection compared to hybridization technology such as microarray, we conducted further analysis on the levels of gene expression changes to enable more stringent and accurate transcriptomic analyses for future studies. By comparing these results to a previous study that identified 11 influenza-specific signatures, we verified that these 11 genes were all expressed in infected nasal epithelium later at 24 hpi, but not at 8 hpi. Furthermore, at both 24 and 48 hpi, all but one of the 11 gene signatures exhibited elevated expression of >2.5-fold change (>1.33 Log_2_FC) compared to uninfected control hNECs (Table 4). When we applied the higher fold change cutoff, the number of significant genes decreased by approximately 50% (Figure 4A), which was also congruent with the linear association observed earlier. Therefore, future studies on early transcriptional alterations could consider adopting the 2.5-fold change in expression as a more stringent threshold, which may be more feasible, especially for large transcriptomic studies that yield large numbers of data points.

In addition, when compared to the previous microarray study on a similar hNEC model [6], both RNAseq and microarray shared a high degree of overlap, with about one third and half of total genes from RNAseq and microarray overlapping, respectively (Figure 4B). The overlap was generally observed in genes with highly altered expression, such as *CXCL10*, *CXCL11*, and *RSAD2*, which were changed to a similar magnitude in both RNAseq and microarray (Appendix A). When we compared the 11 influenza signature genes, RNAseq revealed a more consistent increase in magnitude, i.e., at 48 hpi, the magnitude of the gene change was generally higher than that of the microarray (Table 5). In addition, RNAseq was also able to detect novel genes with expression changes of high magnitude that were generally higher than those found by microarray only (193 genes versus 22 genes with elevated expression greater than 2.5-fold). Genes such as *HEATR9*, *PDCD1*, *IL4I1*, *ART3*, and *KCNH7* were altered to a higher magnitude than the 2.5-fold threshold. Hence, RNAseq-based transcriptomic analysis may augment transcriptomic findings to identify novel gene responses against influenza in the future.

### 3.6. Secretory Cytokine Profiles During Influenza Virus Infection of hNECs Reveal Early Suppression of Cytokines in Apical Secretion

After deriving the transcriptomes by RNAseq, we then further investigated whether the changes in expression of genes resulted in alterations in secretory cytokines and chemokines early in the infection of hNECs. Initially, we detected significant reductions in multiple cytokines at 8 hpi, with the exception of IL-15 which was increased (Appendix A). This may reflect the initial immune suppression during influenza infection. However, at 24 and 48 hpi, less significant changes were observed, i.e., only increase in TNF-a and decrease in MDC and PDGF-AA were noted at 24 hpi. This was followed by increase in IP-10 (CXCL10) and TGF-a and decrease in PDGF-AA seen at 48 hpi. This analysis highlights changes in IP-10, TGF-a, and PDGF-AA to be significant early responses in secretory cytokines/chemokines following influenza infection.

## 4. Discussion

Our study has identified epithelium-initiated host responses which are found to be involved in both innate and adaptive responses. The finding is significant as we can now focus on the primary point of contact of influenza—the nasal epithelium in the study of early host responses for identifying host factors that can be utilized for diagnostic and therapeutic purposes [7]. In addition, our study also showed that it is important for reference databases to use relevant human models like the hNECs model, which contains the mucociliary component of the airways, in order to provide closely representative host responses. While there exists a high number of microarray studies that showed the host responses using similar hNEC models, there are only a small number of equivalent RNAseq studies. Compared to microarrays, RNAseq analysis can provide a more comprehensive picture of the transcriptomic landscape, and is not limited by the hybrid library variant and concentrations [14]. Hence, in order to derive accurate magnitude of gene expression changes, we performed an RNAseq analysis of H3N2 infection using the hNECs model. H3N2 influenza virus was selected, given that it is a major circulating subtype over long periods of time. In addition, relatively lower efficacy of vaccines against this subtype prompted us to study its interactions with the primary host target to elucidate the immune responses and association with adaptive immunity [17,18]. This model has been previously evaluated to be a highly clinically-relevant model that can facilitate controlled infection of nasal cells from multiple individuals. In addition, we have also previously shown—by microarray analysis—that the nasal epithelium is responsible for the initiation of host responses following influenza infection [6]. This renders the hNECs to serve as a valuable tool to analyze transcriptomics from different individuals infected under the same conditions to ensure consistent and relevant responses in humans.

Once the magnitude of gene expression changes was considered, several interesting findings emerged. Firstly, the infected hNECs were observed with strong activation of antiviral genes and early inflammatory genes leading to type I immune responses. A large number of gene expression changes were of magnitude of over 100-fold difference (Log_2_7 to Log_2_9 fold change). Most of the genes with high-magnitude expression changes were verified by qPCR, with statistical significance congruent with the RNAseq analysis. Secondly, despite the absence of immune cells, the infected hNECs were able to generate strong type I responses that may likely aid the recruitment of cytotoxic cells to clear the infected cells. Thirdly, in early responses of the hNECs, IFNλ genes, which represent type III interferons, were more strongly induced than the more frequently observed type I interferons (IFNα and IFNβ), while type II interferons were not produced by hNECs, in agreement with previous studies [5,19]. The induction of type III interferons may reflect an important event within the hNECs where IFNλ, the initial responders against the infection, may be more beneficial in the antiviral response [20,21]. Moreover, we also observed notable suppression of expression of certain genes following influenza infection, including suppression of proliferation and DNA repair genes, which may contribute to the pathology and pathogenesis of influenza [22]. Finally, RNAseq also unraveled expression changes of certain newly-discovered genes in response to influenza infection of the upper airway cells. Genes such as *HEATR9* [23], *IL4I1* [24], *TNFSF13B* (*BAFF*) [25], and *PDCD1* (*PD-1*) [26] are recently implicated in influenza pathogenesis and mucosal defense, thereby signifying the role of the nasal epithelium against influenza infection. Furthermore, RNAseq identified altered expression of *ART3* and *KCNH7* genes that were not previously detected in influenza transcriptomes. These findings hence further reiterate the value of RNAseq in enhancing data on influenza transcriptomes for reference in future studies.

Via pathway enrichment analysis, we have identified known antiviral pathways to validate the hNECs responses against influenza. In addition, we have also documented the potential pathways initiated by the nasal epithelium that may contribute to influenza pathogenesis as represented by the gene expression changes listed in Table 2 and Table 3. By analyses using literature-inferred GO and REACTOME databases, we have demonstrated that the nasal epithelium can play a role in the main antiviral signaling, i.e., IFNγ responses despite not being a direct producer of IFNγ. The pathway enrichment indicated that hNECs may serve as important regulators of type II interferons. Even though the effects of IFNγ are vital to the robust clearance of influenza viruses [27], there are reports of unregulated IFNγ being a contributor to inflammatory damage [28,29]. Therefore, the over-production of IFNγ response factors such as *ICAM1* and *CD44* may contribute to inflammatory damage of the epithelium. Hence, production of factors such as STAT1 [28] by the hNECs is also crucial in ensuring appropriate regulation of IFNγ-mediated expression of influenza response genes to modulate inflammation and to minimize damage. The primary contact of influenza virus with the nasal epithelium may subsequently lead to damage to the airway epithelium as well. This is apparent with the clear enrichment of the pathways of apoptosis, mitochondrial apoptotic processes, and necroptosis that contribute to cell death and mechanical barrier loss during infection [5,30]. Genes such as *IFI6, BAK1, CAPS8, TNFSF10*, and *FAS* suggest active apoptotic cell death that not only destroys cells in the epithelial barrier, but may also serve to propagate the virus and to perpetuate the damage [31,32,33]. Furthermore, during virus infection, aberrant regulation of apoptosis may also lead to further injury to the epithelium and surrounding tissues [34]. On the other hand, necroptosis pathways have also been observed to be enriched in influenza-infected hNECs. Compared to apoptosis, the study of necroptosis in influenza infection is relatively new with contradicting findings [34]. RIPK3/necroptosis studies appear to generate contradictory results as to whether necroptosis protects against or is detrimental during influenza infection [35,36]. Hence, its increased expression during infection of hNECs warrants further investigation on its role in influenza-induced damage. In addition, we also noted enrichment of B-cell signaling pathways by the infected hNECs which may be vital for B-cell responses during the adaptive immune response [37]. We noted that most genes enriched in the B-cell pathways were related to antigen recognition such as proteasome subunits (*PSME2*, *PSMB9*, *PSMA6*, etc.) and B-cell receptor-associated genes such as *DAPP1* and *CARD11* [38]. However, changes in expression of certain growth factors (including *EREG* and *FGF*s) following influenza infection may lead to complications involving airway remodeling and recruitment [39,40]. Further, the effects of the growth factors were further confirmed by the enrichment of pathways related to the proliferation of smooth muscle cells also induced by the infection. Changes to airway smooth muscle cells are usually implicated in airway remodeling [41,42,43], and may also contribute to post-influenza complications. Hence, the genes found in this study may be crucial for elucidating the nasal-initiated responses that may contribute to the pathology and pathogenesis of influenza infection of the airways. Finally, another interesting pathway that may contribute to epithelial damage is the negative regulation of innate immune responses. These genes may serve as proviral factors and aid in immune evasion. For example, *ADAR* is a proviral factor that works in synergy with influenza NS1 to enhance viral replication [44]. *TRAFD1* is a negative regulator of toll-like receptor signaling which is upregulated in influenza-infected hNECs [45]. *DHX58* is a negative regulator of RIG-I/MDA5 signaling pathway [46]. *CEACAM1* is involved in regulation of liver inflammation [47] and its expression appears to exert antiviral effects on influenza virus [48]. *NMI* binds to influenza virus NS1 and inhibits IRF7-mediated interferon signaling [49,50]. Therefore, aberrant expression of genes in this signaling pathway may directly contribute to immune evasion of influenza, culminating in viral propagation and increased epithelial damage.

We summarized the identified pathways (listed in Table 2 and Table 3) that alluded to immune evasion (negative regulation of innate immune responses), antigen processing (metabolism of amino acids and derivatives), and immunomodulation (interferon gamma signaling, B cell receptor signaling, and response to fatty acid) that may contribute to severity of influenza. There was evidence of direct pathway enrichment of potential influenza evasion strategies and/or immunomodulation with accompanying transcriptomic changes. The genes in the pathway may be analyzed for their immunomodulatory activity and whether their expression is beneficial to the virus (immune evasion) or the host (preventing cytokine storm). In addition, the infected hNECs also revealed modified responses associated with fatty acid, with many lipid signaling molecules such as leukotrienes that mediate antiviral responses and subsequent inflammation of the airway [51,52]. Such modified responses may also determine the afforded in the airway and the severity of airway inflammation and damage. In addition, the modification may also affect the lower airway responses to inflammatory mediators; hence, the changes in these pathways may also suggest a potential mechanistic link to the pathogenesis of viral-induced exacerbation of chronic inflammatory diseases. Lastly, we also noted enrichment of pathways related to amino acid metabolism, which is important in antigen processing and proteasomal degradation of foreign protein. The changes in these genes at the hNECs, the target site of influenza infection, may determine the effectiveness of antiviral responses mounted and may therefore influenza disease severity.

In addition, we also compared our RNAseq analysis against previously reported influenza-specific signatures in order to improve future transcriptomic analysis [53]. In vitro transcriptomic analysis yields a large number of differentially-expressed genes that would require additional criteria to identify functionally significant genes. By means of this comparison, we discovered that almost all influenza-specific signatures exhibited differences in expression of above 2.5-fold. Hence, we propose applying fold change of >2.5 as a threshold for future in vitro transcriptomic systems analyses, in order to increase the stringency in detecting functionally significant gene changes.

Finally, we observed that, unlike the transcriptome, there were notably fewer cytokines that were readily secreted during the acute phase of infection. Expression of cytokines was reduced at 8 hpi, except for IL-15, which interestingly is implicated in influenza-induced acute lung injury [54]. Even fewer cytokines showed altered expression at later time points. Among them, only TGF-a, IP-10 (CXCL10), and PDGF-AA were significantly altered at 24 and 48 hpi. These may be significant markers that can be detected in the secretion of influenza-infected mucosal surface that may influence the severity of influenza. IP-10 is a well-established IFNγ response gene, and serves as a useful marker for response against influenza [5,6]. TGF-a represents an important factor involved in the secretion of IL-8 in response to influenza, and may determine the early appropriate innate responses to prevent severe disease [55]. On the other hand, it is also involved in pulmonary fibrosis as a ligand of epidermal growth factor receptor (EGFR) and may contribute to complications in the lower airway [56]. PDGF-AA was found to be elevated in the cerebrospinal fluid of influenza-associated encephalopathy [57], but was consistently reduced in hNEC secretory fluid, thus warranting further investigation into its role in the infected nasal mucosa.

The establishment of a reference transcriptome based on early responses of the human nasal epithelium model serves a key role in research on critical host factors involved in influenza. As the primary host contact with the virus, not only are immune responses against influenza important, but also the alterations in non-immune functions such as metabolism, cell content, and cell cycle, which may contribute to disease severity. In terms of translational potential, the model system identified gene expression changes of significant magnitude and pathways that impact responses against influenza and its severity. These genes may represent novel targets for future diagnostic and therapeutic development. Under controlled conditions, the hNECs clinically establish the baseline for “normal” innate immune responses of the nasal epithelium against influenza viral infection. Such a baseline can be particularly crucial when studying the changes in innate immune responses against influenza, especially in patients with underlying chronic diseases who may have aberrant airway responses against influenza. Their antiviral responses may differ from “normal” subjects, and this study thus provides the basis for comparing the differential responses that culminate in more severe infections in patients with co-morbidities such as diabetes and chronic airway inflammatory diseases. Such comparative clinical studies can potentially enhance the management of influenza viral infection in patients with chronic diseases.

## 5. Conclusions

In conclusion, RNAseq technology allowed us to accurately quantify the magnitude of gene expression changes, as well as the relevant enriched pathways during H3N2 influenza virus infection of hNECs, which can serve as a baseline for future clinical studies. The establishment of this baseline under controlled condition elucidated the antiviral innate response by the infected nasal epithelium, and highlighted the molecular factors and abnormalities in the upper airway that may contribute to influenza severity. Furthermore, this study also culminated in the identification of novel gene signatures and host factors that may be harnessed for future research to develop influenza diagnostic markers and therapeutic targets.

## Figures and Tables

**Figure 1 cells-08-00986-f001:**
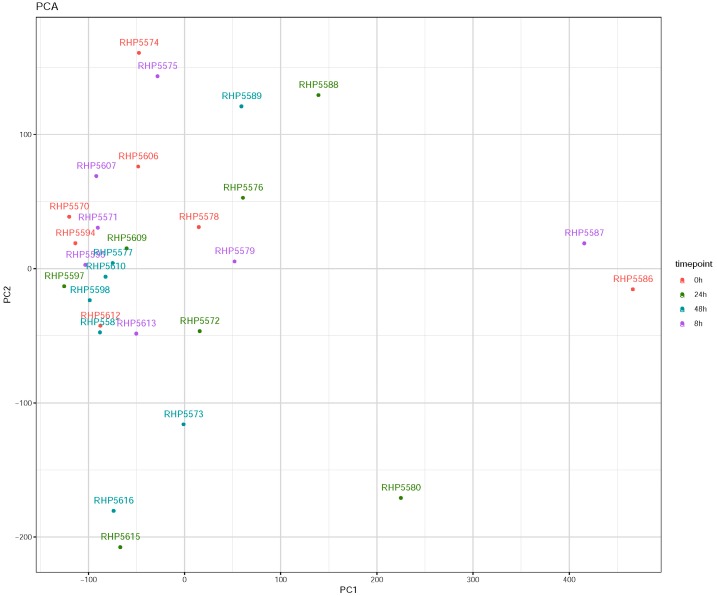
Principal component analysis (PCA) of hNECs from seven donors infected with influenza H3N2 virus. PCA plot of transcriptomic changes of different donor hNECs infected with influenza virus across four time points (0, 8, 24, and 48 h post-infection (hpi)). Each colored dot represents the sequencing ID of infected hNECs from a specific donor at a specific time point of infection. The majority of dots clustered closely following infection, thus signifying the consistency of infection for transcriptomic analysis.

**Figure 2 cells-08-00986-f002:**
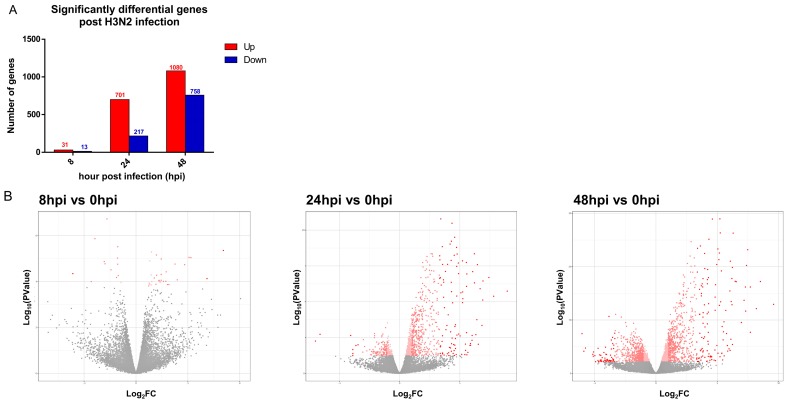
RNAseq analysis of significant gene expression changes following influenza infection. (**A**) Significant gene expression changes over time (FDR < 0.05). Red bars indicate the number of upregulated genes, while the blue bars indicate the number of downregulated genes. The number of significant gene expression changes peaked at 48 hpi. (**B**) Volcano plot showing individual points of gene expression changes over time (8, 24, and 48 hpi). The number of genes decreased in a linear fashion as the fold change increased. Genes with >10-fold difference in expression are depicted in dark red, and shown to include ~10% of the total number of significant gene changes.

**Figure 3 cells-08-00986-f003:**
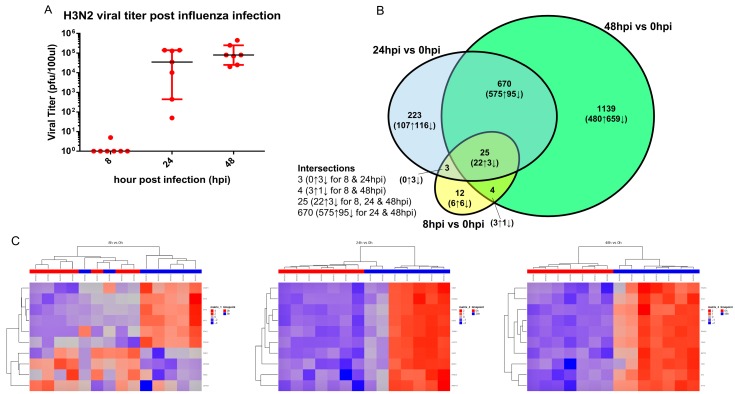
Viral titer and the transcriptomic change pattern post influenza H3N2 infection. (**A**) Viral titer retrieved from apical wash of infected hNECs 8, 24, and 48 hpi of influenza inoculation. The y-axis is presented in a Log_10_ scale and viral titer is presented in PFU/100 µL. Each individual point at each time point represents viral titer retrieved from in vitro-differentiated culture of a hNECs donor. (**B**) Venn diagram of transcriptional change pattern over time post influenza infection. Venn diagram summarizes the transcriptional pattern changes over time. For 8 and 24 hpi, about two thirds of genes overlapped with other time points, while for 48 hpi, one third of genes overlapped. The direction of the gene change in overlapping regions were consistent across the time points it is significant in. A greater number of genes were upregulated than downregulated at all time points. (**C**) Heatmaps of top 10 genes with smallest *p*-value at time points 8, 24 and 48 hpi (blue bar) compared to 0 hpi (red bar). The changes of the genes became more consistent overtime and showed same directional changes and clustering at 48 hpi.

**Figure 4 cells-08-00986-f004:**
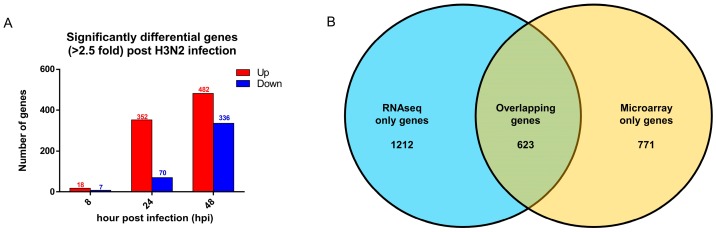
Significant gene expression changes (>2.5-fold change) following influenza infection. (**A**) Significant gene expression changes over time (FDR < 0.05) at >2.5-fold change. Red bars indicate the number of upregulated genes, while blue bars indicate the number of downregulated genes. The number of significant gene expression changes peaked at 48 hpi, with approximately half of the genes being retained following the application of the new threshold. (**B**) Venn diagram of significant genes found in RNAseq (blue) and microarray (yellow). A high degree of overlap was observed where 1/3 of RNAseq genes overlapped with half of the microarray genes.

**Table 1 cells-08-00986-t001:** Summary of RNAseq analysis across time points.

Time, Hours Post Infection (hpi)	8	24	48
**Number of significant genes**	44 (31↑; 13↓)	921 (704↑; 217↓)	1838 (1080↑; 758↓)
**Number of significant genes over 10-fold (>Log_2_FC of 3.4)**	3 (2↑; 1↓)	99 (91↑; 8↓)	167 (112↑; 55↓)
**Highest upregulated gene**	*IFNL2*	*IFNL1*	*IFNL1*
**Highest downregulated gene**	*KLK12*	*NAP1L3*	*CYP26A1*

**Table 2 cells-08-00986-t002:** 48 hpi GO pathways potentially associated with influenza pathogenesis.

GOid	Description	Annotated	Significant	Expected	*p*-Value_adj	Genes
GO:0060333	Interferon-gamma-mediated signaling pathway	72	36	10.35	9.184 × 10^−11^	*TRIM22 ↑; IRF9 ↑; TRIM38 ↑; GBP1 ↑; GBP2 ↑; HCK ↓; HLA-A ↑; HLA-B ↑; HLA-C ↑; HLA-E ↑; HLA-F ↑; HLA-G ↑; ICAM1 ↑; IRF1 ↑; IRF2 ↑; IRF6 ↑; IRF7 ↑; JAK2 ↑; MT2A ↑; OAS1 ↑; OAS2 ↑; OAS3 ↑; PML ↑; B2M ↑; PTAFR ↑; SP100 ↑; TRIM21 ↑; STAT1 ↑; TRIM25 ↑; TRIM26 ↑; CAMK2D ↑; NLRC5 ↑; TRIM5 ↑; OASL ↑; NMI ↑; CD44 ↑*
GO:0008637	Apoptotic mitochondrial changes	81	22	11.64	0.015	*CLU ↓; ERBB4 ↓; EYA2 ↓; IFI6 ↑; GCLC ↓; GCLM ↓; SFN ↑; SLC25A4 ↓; HK2 ↑; IFIT2 ↑; MCL1 ↑; MMP9 ↑; PLAUR ↑; PMAIP1 ↑; BAK1 ↑; SOD2 ↑; BNIP3L ↓; PLA2G6 ↓; CASP8 ↑; AIFM2 ↑; TNFSF10 ↑; NOL3 ↓*
GO:0048661	Positive regulation of smooth muscle cell proliferation	58	19	8.34	0.036	*NAMPT ↑; ABCC4 ↓; PPARGC1A ↓; HBEGF ↑; EREG ↑; FGFR2 ↓; ID2 ↓; IRAK1 ↑; JAK2 ↑; MMP9 ↑; NOTCH3 ↓; IRAK4 ↑; SERPINF2 ↓; PTAFR ↑; PTGS2 ↑; CCL5 ↑; STAT1 ↑; THBS1 ↑; CAMK2D ↑*
GO:0045824	Negative regulation of innate immune response	28	13	4.02	0.015	*ADAR ↑; TRAFD1 ↑; A2M ↓; HLA-E ↑; IFI16 ↑; LGALS9 ↑; SERPINB9 ↑; CEACAM1 ↑; SERPING1 ↑; TNFAIP3 ↑; DHX58 ↑; NLRC5 ↑; NMI ↑*
GO:0070542	Response to fatty acid	32	13	4.6	0.015	*PPARGC1A ↓; CREB1 ↑; CTGF ↓; ALAD ↓; GIPR ↓; PDK4 ↓; PTAFR ↑; PTGS2 ↑; TLR2 ↑; UCP2 ↓; ZC3H12A ↑; CAT ↓; CD36 ↓*
GO:0060544	Regulation of necroptotic process	11	7	1.58	0.043	*RBCK1 ↑; RIPK3 ↑; SLC25A4 ↓; BIRC2 ↑; BIRC3 ↑; CFLAR ↑; ARHGEF2 ↑*

**Table 3 cells-08-00986-t003:** 48 hpi REACTOME pathways potentially associated with influenza pathogenesis.

ID	Description	Gene Ratio	Bg Ratio	*p*-value	*p*.adjust	*q*-value	geneID
877300	Interferon gamma signaling	33/776	65/4637	2.081 × 10^−10^	3.83 × 10^−8^	3.32 × 10^−8^	*OAS2 ↑; OAS3 ↑;IRF7 ↑; OAS1 ↑; STAT1 ↑; TRIM21 ↑; TRIM5 ↑; PML ↑; HLA-F ↑; HLA-E ↑; SP100 ↑; OASL ↑; TRIM22 ↑; TRIM25 ↑; HLA-B ↑; GBP1 ↑; HLA-C ↑; B2M ↑; TRIM38 ↑; IRF6 ↑; HLA-A ↑; IRF9 ↑; JAK2 ↑; TRIM26 ↑; IRF1 ↑; GBP2 ↑; HLA-G ↑; IRF2 ↑; CAMK2D ↑; PTAFR ↑; CD44 ↑; MT2A ↑; ICAM1 ↑*
983705	Signaling by the B Cell Receptor (BCR)	38/776	119/4637	2.72153 × 10^−5^	0.001440597	0.001249039	*PSME2 ↑; BTC ↑; PSMB9 ↑; DAPP1 ↑; PSMB8 ↑; RASGRP3 ↑; RICTOR ↑; EREG ↑; PSMA6 ↑; PSMA4 ↑; MOV10 ↑; HBEGF ↑; ITPR3 ↑; PSMF1 ↑; FGF2 ↑; CARD11 ↑; PSME1 ↑; CREB1 ↑; LYN ↑; PSMB7 ↑; FGFR3 ↓; MALT1 ↑; CDKN1A ↑; PIK3AP1 ↑; AGO1 ↓; FGF1 ↓; FGFR2 ↓; FGF5 ↑; ERBB4 ↓; ITPR1 ↓; UBC ↑; NCK1 ↑; ERBB2 ↓; SH3KBP1 ↑; PSMA3 ↑; PSMA5 ↑; TRIB3 ↑; GAB1 ↑*
109581	Apoptosis	30/776	88/4637	4.88082 × 10^−5^	0.00227963	0.001976505	*PSME2 ↑; PSMB9 ↑; PMAIP1 ↑; TNFSF10 ↑; DSG3 ↑; PSMB8 ↑; BAK1 ↑; TICAM1 ↑; H1F0 ↑; CASP7 ↑; TLR3 ↑; CFLAR ↑; FAS ↑; PSMA6 ↑; PSMA4 ↑; PSMF1 ↑; OCLN ↑; TRAF2 ↑; XIAP ↑; PSME1 ↑; CASP8 ↑; MAGED1 ↓; SFN ↑; PSMB7 ↑; TJP1 ↑; BIRC2 ↑; UBC ↑; PLEC ↑; PSMA3 ↑; PSMA5 ↑*
71291	Metabolism of amino acids and derivatives	40/776	134/4637	9.37168 × 10^−5^	0.004025804	0.003490489	*IL4I1 ↑; IDO1 ↑; PSME2 ↑; SQRDL ↑; PSMB9 ↑; PSMB8 ↑; GAMT ↓; PSMA6 ↑; SMS ↑; PSMA4 ↑; KYNU ↑; SLC5A5 ↓; PSMF1 ↑; SHMT1 ↓; ALDH7A1 ↓; AZIN2 ↑; PSME1 ↑; AFMID ↓; BBOX1 ↓; GCLC ↓; IYD ↓; ALDH4A1 ↓; FAH ↓; PSMB7 ↑; HIBCH ↓; MCCC1 ↓; ALDH18A1 ↓; TST ↓; GCLM ↓; BCAT1 ↑; PSAT1 ↑; GPT ↓; ASNS ↑; PSMA3 ↑; ADI1 ↓; PSMA5 ↑; SLC25A10 ↓; ALDH6A1 ↓*
1168372	Downstream signaling events of B Cell Receptor (BCR)	31/776	98/4637	0.000180538	0.004865597	0.004218614	*PSME2 ↑; BTC ↑; PSMB9 ↑; PSMB8 ↑; RASGRP3 ↑; RICTOR ↑; EREG ↑; PSMA6 ↑; PSMA4 ↑; MOV10 ↑; HBEGF ↑; PSMF1 ↑; FGF2 ↑; CARD11 ↑; PSME1 ↑; CREB1 ↑; PSMB7 ↑; FGFR3 ↓; MALT1 ↑; CDKN1A ↑; AGO1 ↓; FGF1 ↓; FGFR2 ↓; FGF5 ↑; ERBB4 ↓; UBC ↑; ERBB2 ↓; PSMA3 ↑; PSMA5 ↑; TRIB3 ↑; GAB1 ↑*

**Table 4 cells-08-00986-t004:** Summary of alterations in expression of influenza-specific gene signatures in infected hNECs.

Gene_Name	logCPM	FDR 8 h	LogFC 8 h	FDR 24 h	LogFC 24 h	FDR 48 h	LogFC 48 h
***ZBP1***	5.467096	N.S.		2.71 × 10^−6^	6.419275	2.70 × 10^−8^	6.97086
***IFI6***	7.740724	N.S.		2.77 × 10^−14^	4.780717	1.80 × 10^−23^	6.30199
***CD38***	5.090161	N.S.		1.91 × 10^−8^	5.232278	2.10 × 10^−10^	5.699259
***MX1***	10.7197	N.S.		2.36 × 10^−13^	4.87848	3.48 × 10^−17^	5.339009
***HERC5***	6.743117	N.S.		1.05 × 10^−9^	5.093646	6.89 × 10^−12^	5.28321
***IFIH1***	7.876294	N.S.		2.71 × 10^−4^	3.566556	7.51 × 10^−6^	3.972719
***HERC6***	8.376077	N.S.		1.66 × 10^−11^	3.339444	7.15 × 10^−6^	3.843258
***RTP4***	4.998082	N.S.		4.97 × 10^−11^	2.846709	7.28 × 10^−15^	3.25471
***PARP12***	5.511677	N.S.		3.12 × 10^−2^	2.203849	1.99 × 10^−2^	2.11728
***LY6E***	8.624757	N.S.		2.10 × 10^−6^	1.451684	8.74 × 10^−12^	1.989543
***LGALS3BP***	11.0223	N.S.		1.30 × 10^−3^	0.855231	1.57 × 10^−8^	1.336304

**Table 5 cells-08-00986-t005:** Expression of influenza signature genes in infected hNECs at 48 hpi detected by H3N2 microarray and RNAseq.

Gene name	Microarray LogFC 48 h	Microarray FDR 48 h	RNAseq LogFC 48 h	RNAseq FDR 48 h
***ZBP1***	1.30628	1.04 × 10^−6^	6.97086	2.70 × 10^−8^
***IFI6***	3.78685	6.67 × 10^−6^	6.30199	1.80 × 10^−23^
***CD38***	3.61175	5.88 × 10^−7^	5.699259	2.10 × 10^−10^
***MX1***	3.17584	1.57 × 10^−8^	5.339009	3.48 × 10^−17^
***HERC5***	3.53433	1.65 × 10^−8^	5.28321	6.89 × 10^−12^
***IFIH1***	4.08252	5.01 × 10^−8^	3.972719	7.51 × 10^−6^
***HERC6***	3.10932	9.16 × 10^−8^	3.843258	7.15 × 10^−16^
***RTP4***	2.41668	6.25 × 10^−7^	3.25471	7.28 × 10^−15^
***PARP12***	2.02756	4.75 × 10^−8^	2.11728	1.99 × 10^−2^
***LY6E***	1.11967	6.02 × 10^−5^	1.989543	8.74 × 10^−12^
***LGALS3BP***	0.863641	7.59 × 10^−4^	1.336304	1.57 × 10^−8^

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
