# Peer review of "RNA Sequencing of H3N2 Influenza Virus-Infected Human Nasal Epithelial Cells from Multiple Subjects Reveals Molecular Pathways Associated with Tissue Injury and Complications"

_cells, 2019, doi:10.3390/cells8090986_

Round 1

Reviewer 1 Report

In this study, the authors have conducted transcriptomic analysis in H3N2 influenza A virus infected Human nasal Epithelial cells (hNECs) using NGS method. Interestingly, the authors have previously published an article using similar infection model and time points. The only difference being that transcriptome changes were observed using gene microarray (Kai Sen Tan et al., Front. In Micro., 2018). Therefore, the uniqueness/novelty of the current study is missing, which should have been clearly highlighted.

1.    In general, many transcriptome analysis and other studies have shown that in influenza A virus infection results in activation of pathways leading to cell injury as well as changes in metabolisms, though magnitude may change. Therefore, the authors should clearly, not vaguely, discuss the pathways associated with tissue injury and complications, and their uniqueness in this study to support the title.  

2.    It would be interesting the compare the results of present study with previous study clearly.

3.    Quantitative Real-time PCR should be performed to quantify the genes of interest.

Minor points:

1.    Materials and Methods (2.1). Kindly mention that how many donors were used and replace the word “patients” with “donors”.

2.    Titles of supplementary tables missing in the excel files.

3.    Line 336: Kindly explain that how responses were modified?

Reviewer 2 Report

The study by by Tan et al., provides a comprehensive report on the transcriptomic changes associated with H3N2 infection on a clinically relevant model of human nasal epithelial cells (HNECs). The results are well represented and gives a holistic view of the gene expression changes following influenza infection. The data generated from this study using RNA-seq technique is of broad interest to the scientific community in the field of influenza-virology and/or influenza viral-immunology. The HNEC culture system offers a robust and relevant model system to characterize influenza pathogenesis.

Following are some minor points that needs to be answered/discussed by the authors to improve the overall quality of this study:

1.    Have the authors validated the upregulation of at least a few genes (top hits) from RNA-seq data in terms of protein level using a secondary assay such as ELISA or Western Blot?

2.    What is the reason for choosing H3N2 strain for this study? Have the authors tested any other strain of Influenza in their study? Are these genes unique to H3N2? It is important to discuss the rationale behind the choice of the strain ofinfluenza. 

3.    The authors must cite references when they postulate that genes in table 3 could be linked to influenza immune evasion (line numbers 328-332). Have the hits in table 2 and 3 been linked to immune processes in the literature?

Round 2

Reviewer 1 Report

The authors have satisfactorily addressed the raised queries.